# *Slc4a7* Regulates Retina Development in Zebrafish

**DOI:** 10.3390/ijms25179613

**Published:** 2024-09-05

**Authors:** Youyuan Zhuang, Dandan Li, Cheng Tang, Xinyi Zhao, Ruting Wang, Di Tao, Xiufeng Huang, Xinting Liu

**Affiliations:** 1National Clinical Research Center for Ocular Diseases, Eye Hospital, Wenzhou Medical University, Wenzhou 325027, China; zhuangyy@wmu.edu.cn (Y.Z.); lidandan@wmu.edu.cn (D.L.); chengtang@wmu.edu.cn (C.T.); zxyoptometry@wmu.edu.cn (X.Z.); wangruting2000@wmu.edu.cn (R.W.); taodi1123@wmu.edu.cn (D.T.); 2National Engineering Research Center of Ophthalmology and Optometry, Eye Hospital, Wenzhou Medical University, Wenzhou 325027, China; 3State Key Laboratory of Ophthalmology, Optometry and Visual Science, Eye Hospital, Wenzhou Medical University, Wenzhou 325027, China; 4Zhejiang Provincial Clinical Research Center for Pediatric Disease, The Second Affiliated Hospital and Yuying Children’s Hospital, Wenzhou Medical University, Wenzhou 325027, China

**Keywords:** zebrafish, solute carrier family 4 member 7, retinal degeneration, photoreceptors, RPE

## Abstract

Inherited retinal degenerations (IRDs) are a group of genetic disorders characterized by the progressive degeneration of retinal cells, leading to irreversible vision loss. *SLC4A7* has emerged as a candidate gene associated with IRDs, yet its mechanisms remain largely unknown. This study aims to investigate the role of *slc4a7* in retinal development and its associated molecular pathogenesis in zebrafish. Morpholino oligonucleotide knockdown, CRISPR/Cas9 genome editing, quantitative RT-PCR, eye morphometric measurements, immunofluorescent staining, TUNEL assays, visual motor responses, optokinetic responses, rescue experiments, and bulk RNA sequencing were used to assess the impact of *slc4a7* deficiency on retinal development. Our results demonstrated that the knockdown of *slc4a7* resulted in a dose-dependent reduction in eye axial length, ocular area, and eye-to-body-length ratio. The fluorescence observations showed a significant decrease in immunofluorescence signals from photoreceptors and in mCherry fluorescence from RPE in *slc4a7*-silenced morphants. TUNEL staining uncovered the extensive apoptosis of retinal cells induced by slc4a7 knockdown. Visual behaviors were significantly impaired in the *slc4a7*-deficient larvae. GO and KEGG pathway analyses reveal that differentially expressed genes are predominantly linked to aspects of vision, ion channels, and phototransduction. This study demonstrates that the loss of *slc4a7* in larvae led to profound visual impairments, providing additional insights into the genetic mechanisms predisposing individuals to IRDs caused by *SLC4A7* deficiency.

## 1. Introduction

Inherited retinal degenerations (IRDs) comprise an immensely diverse spectrum of retinal degenerative phenotypes, which play a vital role in the etiology of global blindness. IRDs are characterized by progressive pathological changes in photoreceptor cells that are responsible for absorbing and converting light into electrical signals [1]. To date, more than 300 disease-causing genes have been identified in multiple phenotypes of IRDs (RETNET; https://retnet.org, accessed on 15 July 2024). Among these, variants from specific members of the solute carrier family are crucial for maintaining retinal function [2,3,4,5,6,7], such as solute carrier family 4 member 7 (*SLC4A7*). Bok et al. revealed that the loss of *SLC4A7* in mice led to blindness and auditory impairment as a result of degeneration in sensory receptors in the eye and inner ear [8]. Millo et al. documented the presence of the *SLC4A7*-c.2007dup variant in two individuals suffering from autosomal recessive retinitis pigmentosa (ARRP), causing a complete absence of the protein [9]. Ahn et al. identified a new mutation in *SLC4A7*, causing autosomal recessive progressive rod-cone dystrophy [10], emphasizing the importance of ion homeostasis in photoreceptor function and maintenance. In addition, nucleotide variations within *SLC4A7,* identified in genome-wide association studies, have been linked to conditions such as addiction [11], breast cancer [12], hypertension [13], and the accumulation of environmentally toxic metals in the body [14]. Nevertheless, the role and mechanisms of *SLC4A7* in retinal development remain largely unknown.

*SLC4A7* encodes the electroneutral Na^+^/HCO3^−^ cotransporter NBCn1, primarily localized to the plasma membrane, as indicated by subcellular targeting [15,16]. Immunohistochemistry assay (IHC) analysis in the human and mouse retina revealed robust expression across different retinal layers [9]. This electroneutral sodium- and bicarbonate-dependent cotransporter functions with a 1:1 stoichiometry of Na(+):HCO3(−), playing a crucial role in mediating sodium-dependent bicarbonate transport [17,18]. This transport is important for pH recovery following an acid load and for regulating steady-state pH in vascular smooth muscle cells. Additionally, it has a pivotal function in macrophage acidification, aiding the transportation of bicarbonate into the cytoplasm through the expulsion of net acid [19]. Moreover, it supplies cellular bicarbonate for de novo purine and pyrimidine synthesis and serves as a vital mediator of de novo nucleotide synthesis downstream of mTORC1 signaling in proliferating cells [20].

In the present study, we established a zebrafish model deficient in *slc4a7* using morpholino oligonucleotide (MO)-induced knockdown. We employed a diverse range of techniques to evaluate ocular phenotypes, including a quantitative polymerase chain reaction (qPCR), measurement of eye-related dimensions, ocular immunofluorescence, visual behavior experiments, ocular tissue-specific transcriptome analysis, and rescue experiments. Our primary objective was to investigate the impact of *slc4a7* on the visual development of zebrafish. Collectively, these data substantiate the significant role of *slc4a7* in the development of the zebrafish retina. This study holds great promise for advancing our understanding of the pathogenesis of IRDs and for facilitating further preclinical developments in treatment.

## 2. Results

### 2.1. High Expression of SLC4A7 Is Observed in Both Human and Zebrafish Eyes

A comprehensive study of *SLC4A7* expression in human and zebrafish eyes is still lacking. To elucidate the expression profile of *SLC4A7* in human ocular tissues, we began by analyzing single-cell data to evaluate scaled mean expression values across distinct tissue types. In the human eye, *SLC4A7* demonstrates its peak normalized expression levels within the retina (Figure 1A,B). Considering cellular distribution, *SLC4A7* is most abundantly expressed in rods, cones, and amacrine cells (ACs) within the retina, while displaying the lowest expression in retinal ganglion cells (Figure 1C). In terms of subcellular localization, *SLC4A7* protein is predominantly situated in the plasma membrane (Figure 1D). Subsequently, we assessed *slc4a7* expression in zebrafish during early embryogenesis. The qRT-PCR results demonstrated a gradual increase in *slc4a7* expression from 1 to 6 days post-fertilization (dpf) (Figure 1E). Additionally, qRT-PCR was employed to determine the relative expression levels of *slc4a7* in different organs of zebrafish (Figure 1F). The findings revealed that the expression of *slc4a7* in the eye and brain significantly surpassed that in other tissues, with an approximately five-fold increase (Figure 1F).

### 2.2. Silencing slc4a7 Induces Marked Ocular Changes in Zebrafish

For an in-depth investigation into the role of the *slc4a7* gene in zebrafish eyes, a knockdown model was established using an MO specifically designed to target the splice sites of the *slc4a7* gene. The induced expression alterations were subsequently validated through RT-PCR analysis (Appendix A). Three different concentrations of *slc4a7* MOs (0.50 ng, 0.75 ng, 1.00 ng), together with a standard control MO (1.00 ng), were micro-injected into the yolks of embryos, respectively. Remarkably, both groups subjected to high-concentration knockdown (injected with 0.75 ng and 1.00 ng) manifested distinct microphthalmia phenotypes, characterized by a shortened eye axis, diminished eye area, and a reduced eye-axis-length-to-body-length ratio (Figure 2A). In comparison to zebrafish receiving 1.00 ng ctrl MO, those receiving 1.00 ng of *slc4a7*-MO exhibited a 17.27% decrease in eye axis length, a 32.19% reduction in eye area, and a 21.40% decrease in the eye-axis-length-to-body-length ratio (Figure 2B–E).

### 2.3. Aberrant Photoreceptors, RPE, and Amacrine Cells in slc4a7-Deficient Morphants

In order to delve deeper into the retinal structural alterations in *slc4a7*-deficient morphants, we examined photoreceptor cells (PRCs) using the specific marker recoverin (RCVRN) in the AB zebrafish strain (a commonly used laboratory strain). Morphants treated with 1.00 ng of *slc4a7*-MO exhibited a marked decrease in RCVRN within the photoreceptor cells when contrasted with larvae treated with the control MO (Figure 3 A,E). Moreover, we conducted evaluations on symptoms related to retinal degenerations through various transgenic strains of zebrafish, specifically examining amacrine cells, RPE cells, Müller cells (MCs), and blood vessel endothelial cells (BVECs) (Figure 3B–D). In Tg(gad1b:mCherry) zebrafish with *slc4a7* deficiency, the mCherry signal was significantly decreased in both the amacrine cells and the RPE layer compared to the control group (Figure 3 B,F,G). Moreover, within the Tg(kdrl: mCherry) zebrafish strain, *slc4a7*-deficient morphants exhibited a pronounced reduction in signals from endothelial cells of blood vessels surrounding the photoreceptors (Figure 3C,H). Interestingly, within the Tg(gfap:egfp) strain, we observed a pronounced increase in the eGFP signal within the Müller cell layer in *slc4a7*-deficiency morphants (Figure 3D,I), implying that the knockdown of *slc4a7* damages the retina and induces the activation of Müller cells. 

Using Tg(gad1b:mCherry) transgenic larvae, we conducted additional observations of the optic tectum region to investigate potential developmental alterations in the zebrafish visual center associated with reduced *slc4a7* expression. Live fluorescent imaging comparisons were performed between the control and knockdown groups at 5 dpf (Appendix A). The results demonstrate that *slc4a7* knockdown (1.00 ng) leads to the relative fluorescence intensity of the optic tectum (Appendix A).

### 2.4. Enhanced Apoptosis in Retinas Following slc4a7 Gene Silencing

We conducted TUNEL staining to evaluate whether *slc4a7* knockdown induces cell death in the retina. Embryos injected with standard morpholinos (1.00 ng) displayed minimal apoptotic signals at 5 dpf (Figure 4). However, a noticeable increase in apoptotic cells was evident in the retina of the *slc4a7*-MO 1.00 ng group compared to the control group (Figure 4). 

### 2.5. Impaired Visual Behaviors in Zebrafish with slc4a7 Gene Suppression

To further evaluate the impact of *slc4a7* knockdown on visual function in zebrafish, assessments were conducted using optokinetic response (OKR) and visual motor response (VMR) based on a previously described protocol [2]. Morphants subjected to a 1.00 ng *slc4a7*-MO injection demonstrated diminished activity in both the ON and OFF conditions at 5 dpf (Figure 5A–D). Notably, the impairment in OFF responses (0.023) was significantly more pronounced than that observed in ON responses (0.079) in *slc4a7*-deficient morphants. Moreover, the OKR findings reveal a stepwise reduction in eye movement frequency as the concentration of *slc4a7* increases over the course of one minute (Figure 5E,F). 

### 2.6. RNA Sequencing Analysis Revealed Distinct Transcriptomic Changes in slc4a7-Knockdown Zebrafish Eyes

We then examined *slc4a7*-mediated downstream targets that could account for the aberrant photoreceptors, RPE, and amacrine cells during the pathogenesis of retinal degeneration. RNA sequencing analysis was conducted on the eyeballs of slc4a7-MO and control zebrafish from 5 dpf to understand the regulatory role of slc4a7 in larval eye development (Figure 6A–C). Table 1 provides details on the top 15 upregulated and downregulated genes. Additionally, a total of 3731 differential alternative splicing (DAS) events were identified, primarily involving exon skipping and alternative splicing junctions (Appendix A). Furthermore, the deficiency of *slc4a7* manifested noticeable effects on gene expression as evidenced by the outcomes of gene ontology (GO) enrichment analysis. This analysis identified significantly perturbed categories among genes that were either downregulated or upregulated (Appendix A). In terms of biological processes (Figure 6D), DEGs were significantly enriched in pathways associated with visual perception (GO: 0007601), monoatomic ion transport (GO: 0006811), and homophilic cell adhesion via plasma membrane adhesion molecules (GO: 0007156). At the molecular function level (Figure 6D), the upregulated genes were notably associated with calcium ion binding (GO: 0005509), monoatomic ion channel activity (GO: 0005216), and G-protein-coupled photoreceptor activity (GO: 0008020). Furthermore, the cellular component analysis highlighted a significant presence of upregulated genes in plasma membrane (GO: 0005886), synapse (GO: 0045202) and cell junction (GO: 0030054) (Figure 6D). Collectively, these GO terms predominantly encompass aspects related to vision, ion channels, and neural signal transduction. On the other hand, according to KEGG pathway enrichment analysis (Figure 6E), the differentially expressed genes (DEGs) are predominantly associated with pathways such as phototransduction (Appendix A), melanogenesis, the p53 signaling pathway (Appendix A), neuroactive ligand–receptor interaction (Appendix A), progesterone-mediated oocyte maturation, etc.

### 2.7. The Reversal of Knockdown Effects via mRNA Compensation

To verify that the ocular phenotypes observed in *slc4a7*-deficient morphants were not attributed to off-target effects, we conducted a rescue experiment employing mRNA compensation. Full-length slc4a7 mRNA (200 ng/μL) and slc4a7 MO (1.00 ng per embryo) were co-injected into embryos. We evaluated the phenotypes by measuring ocular size, VMR, and apoptosis staining and fluorescence observations. Remarkably, a pronounced recovery from microphthalmia was evident in the larvae co-injected with mRNA and MO (Figure 7A–D). Moreover, larvae co-injected with mRNA and MO exhibited significantly restored ON and OFF responses (Figure 7D,E). In addition, the apoptosis staining and fluorescence observations of the co-injected group did not exhibit an obvious change compared to the control group (Appendix A).

## 3. Discussion

In this study, we designed and generated *slc4a7* knockdown zebrafish in both AB and transgenic backgrounds to assess their retinal abnormalities. Observations of ocular phenotypes revealed that *slc4a7* knockdown in zebrafish resulted in a shortened eye axial length, decreased eye area, and reduced eye–body ratio. Fluorescence measurements show a marked decrease in immunofluorescence in photoreceptors, as well as in mCherry fluorescence in the RPE and amacrine cells of larvae with *slc4a7* knockdown. Conversely, there is an augmentation in fluorescence signals in Müller cells. Furthermore, the thickness of the vascular network surrounding photoreceptors experiences a noticeable reduction. TUNEL staining revealed the widespread apoptosis of retinal cells caused by *slc4a7* knockdown. Visual behavior experiments revealed a notable decline in or absence of ON and OFF responses in the *slc4a7* knockdown group, accompanied by a substantial reduction in eye movement reactions. Importantly, the disease phenotypes derived from *slc4a7* deficiency were improved by *slc4a7* mRNA compensation. 

Some theories provide potential causes of the impact of slc4a7 knockdown (the electroneutral Na^+^/HCO3^−^ cotransporter) on the visual behavior and retinal development of zebrafish. At the molecular level, Na^+^ and HCO3^−^ ions directly or indirectly modulate membrane conductivity by regulating ion channel activity [21,22], thereby influencing neuronal excitability and signal transduction. The balance of Na^+^ and HCO3^−^ in the retina plays a crucial role in normal visual transduction and the functioning of the visual system. When cone and rod cells are stimulated by light, the photosensitive pigment (visual opsin) undergoes a conformational change [23]. This change in conformation activates ion channels on the rod cell membrane, leading to an influx of primarily calcium ions [24]. The concentrations of Na^+^ and HCO3^−^ are closely associated with Ca2^+^ levels [25,26]. Reduced transport of Na^+^ and HCO3^−^ leads to a decrease in intracellular negative charge, subsequently affecting Ca2^+^ balance [25,26]. Fluctuations in intracellular Ca2^+^ levels can influence cellular signal transduction, apoptosis, and other biological processes [27]. The findings from our GO and KEGG pathway analyses (such as neuron projection, photoreceptor activity, G-protein-coupled photoreceptor activity, calcium ion binding, and regulation of monoatomic ion transmembrane transport) consistently back up the above hypotheses.

The results of transcriptome analysis indicate that numerous differentially expressed genes in the *slc4a7*-knockdown zebrafish eyes were significantly enriched in multiple upregulated pathways, including the p53 signaling pathways, apoptosis, and ECM–receptor interactions. The p53 signaling pathway plays a pivotal role in maintaining the biological homeostasis of cells, encompassing cell cycle regulation [28], DNA repair [29], apoptosis [30], the inhibition of angiogenesis [31], and the preservation of genomic stability [32]. The heightened presence of TUNEL-positive signals in the retina of *slc4a7* morphants is consistent with the activation of both the apoptosis and p53 signaling pathways. Furthermore, we observed a marked decrease in signals emanating from endothelial cells of blood vessels surrounding the photoreceptors in *slc4a7*-deficient morphants, which may be linked to the inhibitory effect of the p53 signaling pathway on the process of angiogenesis. Notably, Müller glia, the primary glial cell type in the retina, have the potential for proliferation and neuronal regeneration following retinal damage [33]. In addition, we cannot yet rule out the potential possibility that the activation of the p53 pathway may be due to non-specific toxicity of the MO. Our fluorescence observations indicate a significant increase in Müller cell signals in *slc4a7* morphants compared to the control group. This finding is consistent with the discovery of Nomura-Komoike et al. that the cell cycle re-entry of rat Müller glia is accompanied by the upregulation of p53 [34].

The results of our investigation provide initial insights into the influence of slc4a7 on visual health and retinal development in zebrafish, highlighting a range of retinal degeneration phenotypes evident between 3 and 5 dpf. However, several study limitations need to be considered. The stability of MO effects diminishes after 7 dpf, preventing us from conducting longer-term observations to assess the role of *slc4a7* in the zebrafish retina. To this end, we also used CRISPR/Cas9 genome editing to create *slc4a7* crispants, aiming for a longer observation window in the future [35]. Similar changes in ocular phenotypes were observed in the mosaic *slc4a7* crispants (Appendix A). Zebrafish possess several advantages, including genetic tractability, external fertilization, large eye–body ratio, and rapid development of their visual system [36]. The retina of zebrafish larvae reaches a developmental stage similar to that of adults by 3 dpf [37]. Moreover, MO knockdown presents benefits such as low time investment, cost-effectiveness in experiments, and simplicity in operation. Combining these strengths will greatly aid in the screening of anti-IRD drugs and advancing the investigation of molecular mechanisms underlying IRDs in the future [38].

## 4. Materials and Methods

### 4.1. Zebrafish Husbandry and Embryo Preparation

Adult zebrafish of the AB wild-type, Tg(gad1b:mCherry) and Tg(kdrl:mCherry) strains were obtained from the China Zebrafish Resource Center (Wuhan, China). The Tg(gfap:egfp) strain zebrafish were sourced from Hunter Biotech in Nanjing, China. All adult zebrafish were housed at a constant temperature of 28.5 °C in a standard circulating aquaculture system. Zebrafish larvae and embryos were cultured at 28.5 °C following the protocol outlined by Kimmel et al. [39]. All experimental procedures strictly adhered to the guidelines established by the Association for Research on Vision and Ophthalmology (ARVO) regarding the ethical use of animals in ophthalmic and vision research. Moreover, all experiments received the necessary approval from the Institutional Animal Care and Use Committee of Wenzhou Medical University. 

### 4.2. Morpholino Knockdown Experiments

The *slc4a7*-targeting MO was designed to target the first splice junction of the *slc4a7* gene, which spans the non-coding region of the first intron and the second exon (5′-CATCATTGCCCTGAATCGAGAGAGA-3′). To validate the efficiency of the MO injections and the knockdown of *slc4a7* expression, rescue experiments, RT-PCR analysis, and Sanger sequencing were conducted (Appendix A). The primer sequences are listed in Appendix A for RT-PCR. The standard non-target control MO (ctrl-MO) sequence was as follows: 5′-CCTCTTACCTCAGTTACAATTTATA-3′ [40]. All MOs were synthesized by Gene-Tools, LLC (Corvallis, OR, USA). The *slc4a7*-targeting MO and the ctrl MO were individually microinjected into the yolks of one-cell-stage embryos at three different doses (0.50 ng, 0.75 ng, and 1.00 ng) [41,42].

### 4.3. Quantitative Real-Time PCR (qRT-PCR)

Tissue-specific qRT-PCR was performed on various organs dissected from 3-month-old wild-type AB zebrafish strain. For time series qRT-PCR, wild-type larval zebrafish (*n* = 18 for each PCR sample) were collected at different time points, ranging from 1 day post-fertilization (dpf) to 6 dpf. Total RNA extraction was carried out using a Vazyme RNA extraction kit (Nanjing, China), followed by reverse transcription with a Vazyme reverse transcription kit (Nanjing, China). Subsequent qRT-PCR analysis was conducted using the ABI Q6 detection system (Applied Biosystems; Thermo Fisher Scientific, Inc., Waltham, MA, USA). β-actin served as the endogenous reference. Relative mRNA expression was determined using the 2^−ΔΔCt^ approach. Each replicate in qRT-PCR was subjected to triplicate testing. Primer sequences are provided in Appendix A. QRT-PCR experiments for Figure 1E,F were biologically repeated three times.

### 4.4. Ocular Measurement

Ocular parameter measurements were carried out following established protocols, as described in a prior publication [42,43]. Each trial involved 6–10 larvae per group, and we performed these experiments in triplicate. Images capturing both the lateral and vertical perspectives of each larva were obtained using a microscopic camera (SZX116, OLYMPUS, Tokyo, Japan). Subsequently, measurements of axial length, eye area, and body length were conducted through the integrated software (OLYMPUS cellsens standard 1.14).

### 4.5. Slc4a7 mRNA Preparation and Rescue Experiments

Complying with the guidelines for morpholino use in zebrafish [44], we cloned the full-length zebrafish slc4a7 cDNA into the pUC57 vector, incorporating the T7 promoter sequence and Kozak sequence at the 5′ end, to rescue the knockdown phenotype observed in slc4a7-deficient morphants. The DNA templates were synthesized directly by Sangon Biotech (Sangon, China). For amplification, we used primers containing the T7 promoter for zebrafish *slc4a7* cDNA: 5′-TAATACGACTCACTATAGGGGCCACCATGGAGG-3′ and 5′-TCATAGAGATGTCTCTGTATCCACACGCTTGTCC-3′. After high-fidelity PCR, the template DNA was purified using the QIAquick PCR Purification Kit (Qiagen, Hilden, Germany). Capped full-length mRNAs were synthesized using the mMESSAGE mMACHINE T7 ULTRA Transcription Kit (Invitrogen, Carlsberg, CA, USA), followed by purification with an RNeasy Mini Kit (Qiagen) according to the manufacturer’s instructions. Subsequently, full-length zebrafish *slc4a7* mRNA (400 ng/µL) and *slc4a7* MO (2.0 ng) were mixed in equal volumes and co-injected into one-cell stage embryos, resulting in a final concentration of 200 ng/µL mRNA and a dose of 1.00 ng MO per embryo.

### 4.6. TUNEL Assay

Apoptosis within the zebrafish retinal tissue was evaluated through the use of the terminal deoxynucleotidyl transferase dUTP nick end labeling (TUNEL) assay. We followed standard procedures with a One-Step TUNEL Assay Kit (Beyotime, Beijing, China) [45]. In brief, the ocular tissues were fixed in 4% paraformaldehyde, washed with PBS, and permeabilized using 0.5% Triton X-100 to expose the fragmented DNA within apoptotic cells. Subsequently, TUNEL detection solution was applied to each sample, followed by a 60-min incubation at 37 °C in the absence of light. Finally, the sections were rinsed with PBS and counterstained with DAPI (4,6-diamidino-2-phenylindole).

### 4.7. Immunohistochemistry

Zebrafish larvae, sourced from either the AB wild-type strain or a transgenic strain, were initially fixed in 4% paraformaldehyde for one hour at 5 dpf. Subsequently, they underwent a gradual dehydration process, involving 15%, 22.5%, and 30% sucrose solutions. These prepared specimens were then embedded in an optimal cutting temperature compound (Tissue Tek, Torrance, CA, USA) and cryo-sectioned longitudinally at a thickness of 12 µm. Additionally, sections from the AB wild-type strain were subjected to blocking with 5% bovine serum albumin (BSA) at room temperature. They were further incubated overnight at 4 °C with an RCVRN antibody (rabbit; 1:1000; proteintech), followed by a 2.5-h incubation with goat anti-rabbit IgG secondary antibodies labeled with Alexa Fluor 488 at room temperature. Finally, all samples were stained with DAPI. The acquisition of images was carried out using a DM4b microscope (Zeiss, Oberkochen, Germany).

### 4.8. Transcriptome Analyses

To assess the impact of *slc4a7* knockdown on the transcriptome, we conducted an RNA-seq analysis. We dissected the eyeballs of 36 zebrafish at 5 dpf in groups of *slc4a7* MO 1.00 ng and ctrl-MO 1.00 ng, respectively. Afterwards, we randomly divided the eyes collected from each group into three equal parts as biological replicates. Total RNA was obtained through the utilization of the Trizol reagent kit (Life Technologies, Carlsbad, CA, USA) in accordance with the manufacturer’s instructions. Subsequently, the concentration of RNA was determined using Nanodrop 2000 spectrophotometers (Thermo Fisher Scientific, Waltham, MA USA), and the integrity of the RNA was assessed by employing the Agilent 2100 Bioanalyzer (Agilent Technologies, Santa Clara, CA, USA). Sequencing libraries were generated using VAHTS Universal V5 RNA-Seq Library Prep Kit for Illumina^®^. RNA sequencing was conducted by OE Biotech Co., Ltd. (Shanghai, China), employing the Illumina Novaseq 6000 platform (Illumina, CA, USA). The quality of data was evaluated using FASTQC 0.18.0. Subsequently, mRNA sequences were aligned to the Danio_rerio.GRCz11.100.genome.fa using HISAT2 (version 2.1.0), and gene counts were derived from the mapping files using StringTie (version 2.1.4). DESeq2 software (version 1.20.0) [46] was used to investigate differential RNA expression across different groups. Genes with a log2 fold change (LFC) greater than 1 and a false discovery rate (FDR) less than 0.05 were considered significantly differentially expressed. Following this, Gene Ontology (GO) terms and Kyoto Encyclopedia of Genes and Genomes (KEGG) pathways were generated using WebGestalt [47] and g:Profiler [48]. ASprofile [49] was utilized to detect the alternative splicing of differentially regulated transcripts isoforms or exons in each sample, while rMATS [50] was used for processing differential alternative splicing analysis.

### 4.9. Optokinetic Response and Visual Motor Response Assay

To investigate vision-guided behavior in zebrafish larvae, we executed an optokinetic response (OKR) analysis following established procedures when the larvae reached 5 dpf. We employed OKR software (ViewPoint OKR 2.0, France) to record the eye movements of the larvae during a 1-min observation period. All behavioral assessments were conducted at the 5 dpf stage [42]. Furthermore, we performed visual motor response (VMR) analysis according to standard procedures at 5 dpf [51]. Data collection involved 12 larvae per experimental group, placed in a 96-well plate, and subjected to 3 h of dark adaptation before the behavioral tests. A ZebraBox (ViewPoint 2.0, Lyon, France) was programmed to provide the larvae with three cycles of ON and OFF light stimuli (each lasting 30 min), while the machine tracked the zebrafish larvae’s activity on a per-second basis.

### 4.10. Subcellular Localization and Expression Analysis of Human SLC4A7

Unified confidence scores for the subcellular localization evidence of SLC4A7 were derived from COMPARTMENTS subcellular localization database (https://compartments.jensenlab.org/, accessed on 1 June 2024). Single-cell data from various human ocular tissues (including the sclera and choroid, retina, iris, and cornea) were obtained from the public dataset GSE147979 [52]. The analysis of the distribution and scaled mean expression values of SLC4A7 was conducted based on this dataset using the online tool Single Cell Portal (https://singlecell.broadinstitute.org/single_cell/study/SCP1311/, accessed on 1 June 2024).

### 4.11. Generation of slc4a7 Crispants

The targeted sequence (TCTGCAGTTCAGGTCCAGCGTGG) of zebrafish slc4a7 was designed using an online tool CHOPCHOP (http://chopchop.cbu.uib.no/, accessed on 15 July 2024) based on the composite score. The scrambled sequence (GCAGGCAAAGAATCCCTGCC) was used to generate sham-injected control larvae [53]. To generate the slc4a7 crispants, we employed a swift CRISPR/Cas9 gene knockout technique as detailed in reference [53]. The templates for inducible RNAs targeting slc4a7 and scrambled regulatory genes were generated through annealing and elongation, followed by in vitro transcription using a T7 High-Yield RNA Transcription Kit (Vazyme Biotech; TR101-01, Beijing, China). The sgRNAs (final concentration of 1 µg/µL) were mixed with Cas9 protein (final concentration 1 µg/µL) (Novoprotein, E365-01A, Summit, NJ, USA) in a Cas9 buffer solution and incubated at 37 °C for 5 min to form a ribonucleoprotein complex (RNP). Microinjection was performed by delivering 1 nL of the mixture into the yolk cells of one-cell-stage embryos.

### 4.12. Quantification and Statistical Analysis

For the assessment of relative expression levels of mCherry, eGFP, and RCVRN, the retinal pigment epithelium (RPE), photoreceptor layer, outer nuclear layer (ONL), and inner plexiform layer (IPL) within zebrafish retinas were manually outlined in Adobe Photoshop without bias. Subsequently, the normalized immunofluorescence intensities along specific anatomical layers were measured within these outlines using R programming (version 4.0.5). To quantitatively analyze the mean fluorescence intensity in the optic cup region, we utilized Fiji software (version 2.4.0) to measure the fluorescence intensity (mean gray value) for the respective areas. Statistical analyses were conducted utilizing Python (version 3.7.3) and R Studio (R version 3.6.1). To compare the two groups, an unpaired Student’s *t*-test was employed. To compare the means among multiple groups, a one-way ANOVA was employed followed by Tukey’s post hoc tests. Bar plots are shown as the mean ± s.e.m. Statistical significance was defined as a *p*-value less than 0.05, denoted as * for *p* < 0.05, ** for *p* < 0.01, and *** for *p* < 0.001.

## 5. Conclusions

In summary, we revealed the regulatory role of *slc4a7* in the retinal development of zebrafish and elucidated the molecular mechanisms underlying *slc4a7*’s regulation of ocular development through ocular-tissue-specific transcriptome analysis. This study highlights the significant potential to enhance our comprehension of the related mechanisms underlying retinal degeneration and to promote additional preclinical advancements in therapeutic approaches.

## Figures and Tables

**Figure 1 ijms-25-09613-f001:**
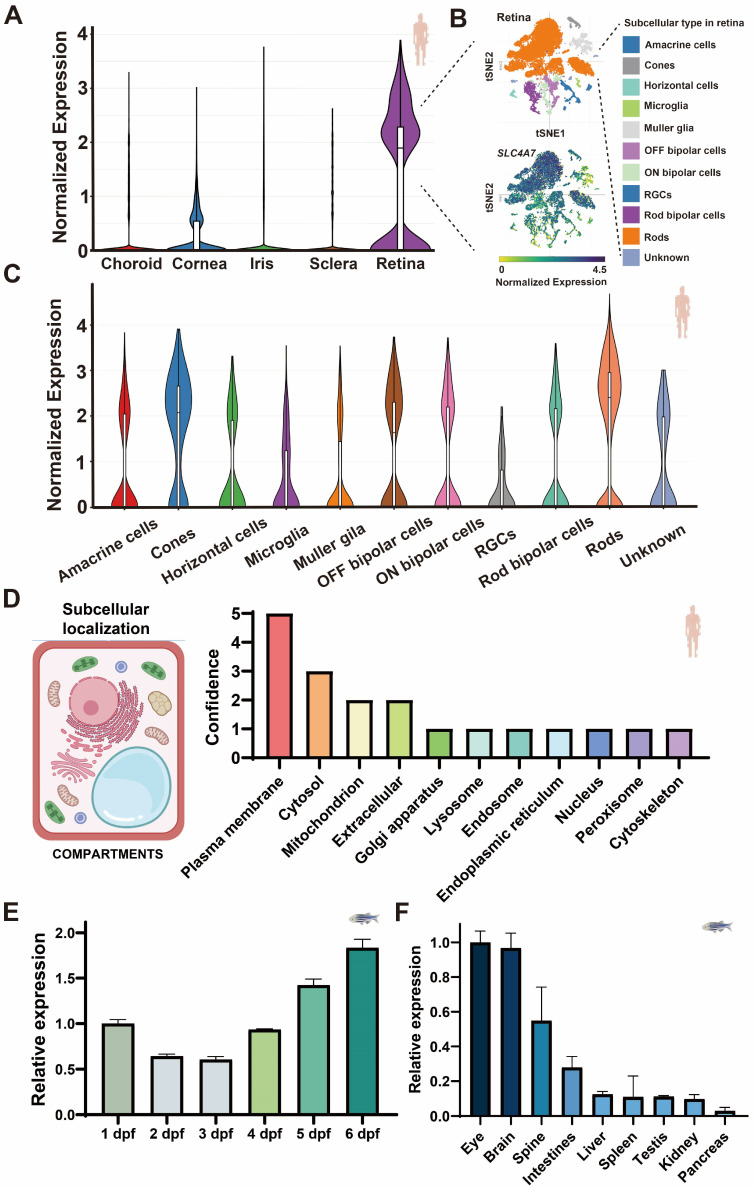
The expression profile of *SLC4A7 (slc4a7*) in both humans and zebrafish. (**A**) Profiling of human single-cell RNA sequencing unveils *SLC4A7* expression in various ocular tissues, including the retina, cornea, iris, sclera, and choroid. (**B**) Single-cell transcriptome profiling and *SLC4A7* gene signatures of the human retina. The upper tSNE plot showing major cell subsets in human retina. The lower tSNE plot of all cells colored by enrichment of *SLC4A7* gene signatures. (**C**) Detailed expression pattern of *SLC4A7* across various cell types in the human retina. (**D**) Human subcellular localization of *SLC4A7* provided by COMPARTMENTS. (**E**,**F**) qRT-PCR display the time series (**E**) and the tissue-specific (**F**) expression pattern of *slc4a7* in zebrafish larvae. Bar plots are shown as the mean ± std. Label of human and zebrafish indicating the species source of the data. The normalized expression values refer to the standardized measure of gene expression levels across different cells or cell types in a single-cell RNA sequencing dataset, as calculated and provided by the online analysis tool, Single Cell Portal. The relative expression values were calculated relative to the expression of the reference gene β-actin. The dashed lines indicate detailed information about the corresponding tissues or figures. Human or zebrafish icons indicate the species origin of the data.

**Figure 2 ijms-25-09613-f002:**
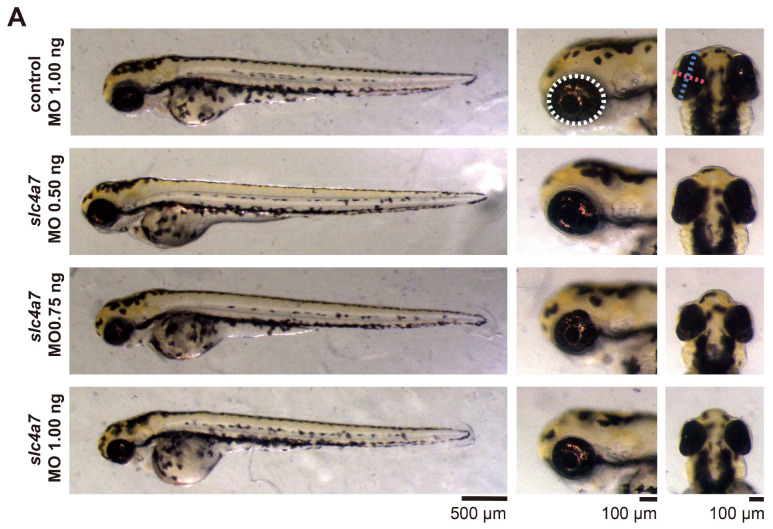
*Slc4a7*-deficient zebrafish morphants exhibited marked ocular changes. (**A**) Whole-body view (scale bar = 500 μm), lateral view (scale bar = 100 μm), and vertical view (scale bar = 100 μm) of zebrafish larvae at 3 dpf. The ocular area is indicated by white circles. The ocular axis is indicated by a red dashed line. The equatorial axis is indicated by a blue dashed line. (**B**–**E**) The measurement of ocular axis length, equatorial length, ocular area, and the ratio of ocular length to body length. *n* = 25 in each group. Bar plots are shown as the mean ± s.e.m. Data were analyzed using one-way ANOVA followed by Tukey’s post hoc tests, *** *p* < 0.001, and **** *p* < 0.0001 indicate significant differences from the control 1.00 ng group. The scattered circles, squares, and triangles of various colors in the bar chart denote individual values of zebrafish eye-related parameters for different groups.

**Figure 3 ijms-25-09613-f003:**
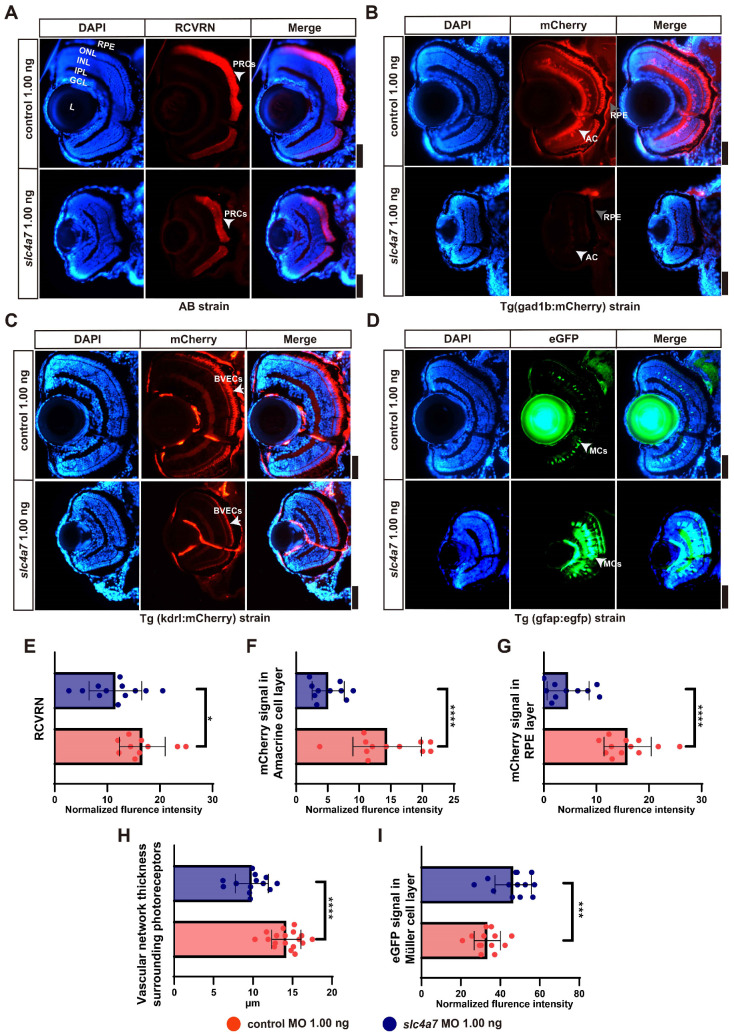
Loss of *slc4a7* leads to retinal abnormalities in zebrafish larvae. (**A**) Immunostaining for RCVRN in *slc4a7*-deficient and control AB zebrafish strains at 5 dpf. (**B**) The fluorescence imaging depicts retinal pigment epithelium cells and amacrine cells in *slc4a7*-deficient and control Tg (gad1b:mCherry) strain at 5 dpf. (**C**) The fluorescence imaging focusing on blood vessel endothelial cells in *slc4a7*-deficient and control Tg(kdrl:mCherry) strains at 5 dpf. (**D**) The fluorescence imaging illustrates Müller cells in *slc4a7*-deficient and control Tg (gfap:egfp) strains at 5 dpf. (**E**) Statistical results for normalized fluorescence intensity of RCVRN. (**F**,**G**) Statistical results for normalized fluorescence intensity of mCherry signal in amacrine cells (ACs), (**F**), and the RPE layer (**G**). (**H**) Statistical results for vascular network thickness surrounding photoreceptors. (**I**) Statistical results for the normalized fluorescence intensity of the eGFP signal in Müller cell layer. Scale bar = 50 μm. Bar plots are shown as the mean ± s.e.m. The *t*-test was performed between the two groups. * *p* < 0.05, *** *p* < 0.001, **** *p* < 0.0001. GCL: ganglion cell layer; IPL: inner plexiform layer; INL: inner nuclear layer; OPL: outer plexiform layer; ONL: outer nuclear layer.

**Figure 4 ijms-25-09613-f004:**
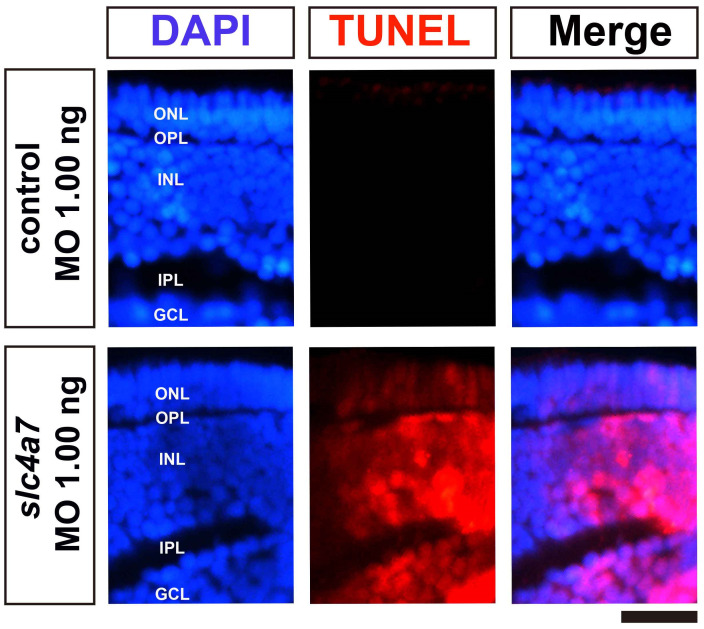
Knockdown of *slc4a7* resulted in a marked elevation of apoptosis in the zebrafish retina. TUNEL assay was used to detect apoptosis in larval retinas at 5 dpf. Knockdown of *slc4a7* led to a marked elevation in the number of TUNEL^+^ cells in the retina at 5 days dpf. Scale bars = 20 μm. Blue indicates cell nucleus stained by DAPI. Red indicates cells with a positive TUNEL reaction.

**Figure 5 ijms-25-09613-f005:**
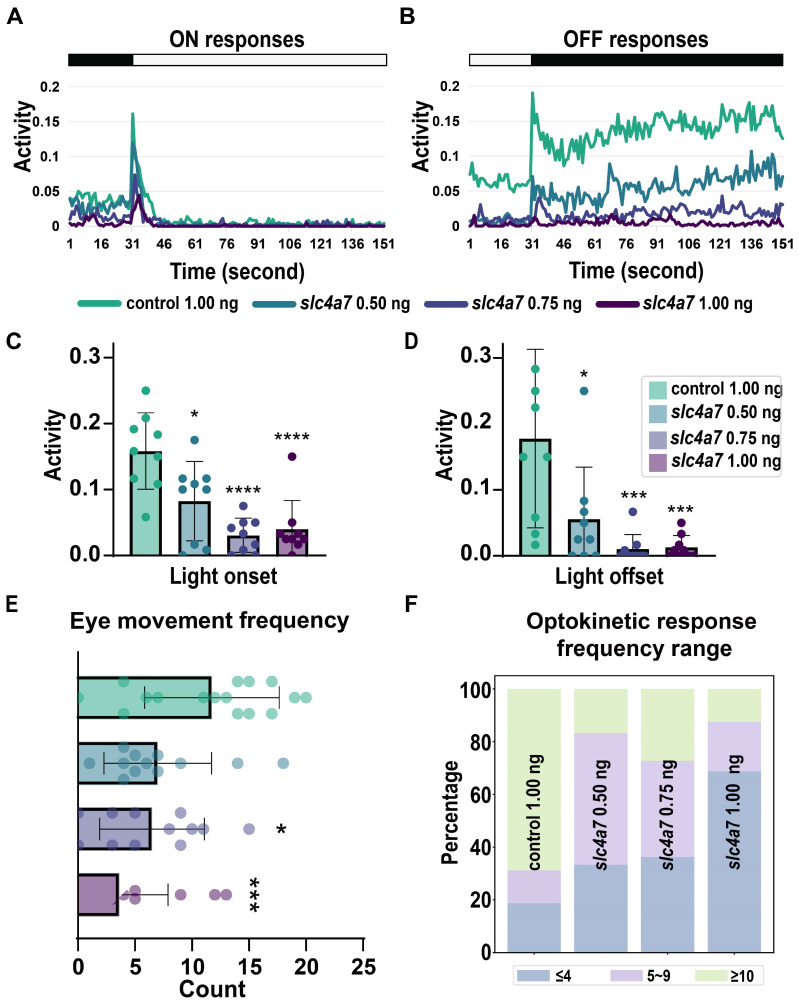
Zebrafish morphants lacking *slc4a7* exhibited impaired visual behaviors. (**A**,**B**) VMR testing results are depicted in the line charts for *slc4a7*-deficient zebrafish morphants. (**C**,**D**) Scatter plots with bar show the quantification of ON responses and OFF responses. Each point represents the average activity of 12 larvae within the group at the moment of each light environment change. Error bars represent standard deviation (STD). (**E**,**F**) Frequency distributions of eye movements (times/minute) in zebrafish larvae are illustrated in the stacked boxes. Data were analyzed using one-way ANOVA with Tukey’s post hoc tests, * *p* < 0.05, *** *p* < 0.001, **** *p* < 0.0001. The scattered circles of various colors in the bar chart denote individual values of zebrafish eye-related parameters for different groups.

**Figure 6 ijms-25-09613-f006:**
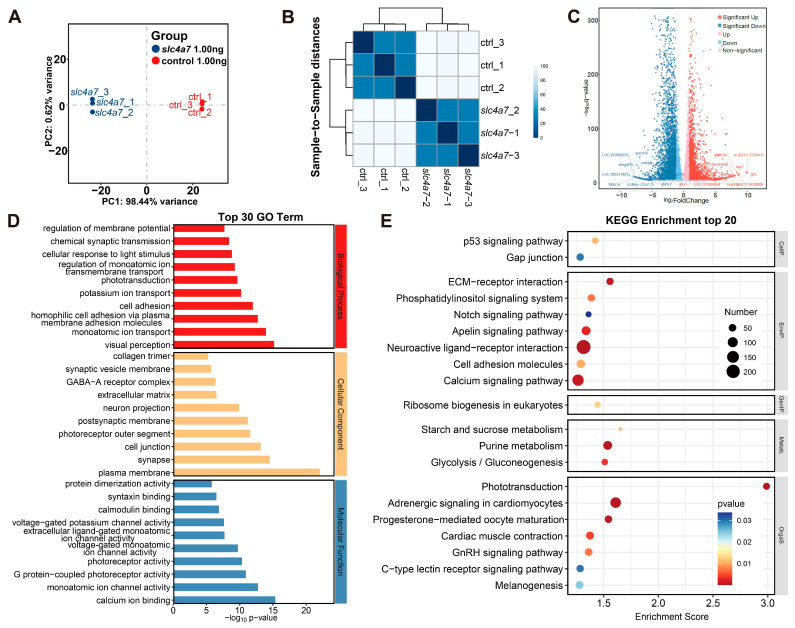
Transcriptional profiling of bulbus oculi from *slc4a7*-deficient zebrafish morphants. (**A**) Principal component analysis (PCA) of the expressed genes showing *slc4a7* and control sample separation. (**B**) Heatmap of sample-to-sample distances. (**C**) Volcano plot showing highlights of DEGs from *slc4a7* KD eyes compared with control eyes. (**D**) GO analysis identified the top 30 most significant GO terms in the *slc4a7* 1.00 ng group compared to the control 1.00 ng group. (**E**) Significantly enriched KEGG pathways (top 20) in the *slc4a7* 1.00 ng group compared to the control 1.00 ng group.

**Figure 7 ijms-25-09613-f007:**
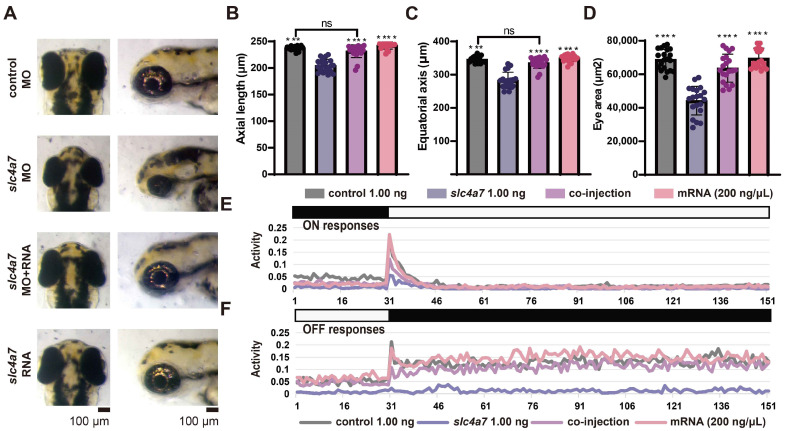
*Slc4a7* mRNA compensation rescues phenotypes in slc4a7-deficient morphants. (**A**) Enlarged vertical and lateral views of larval eyeballs. Larvae injected with slc4a7 MO and full-length slc4a7 mRNA exhibit normal-sized eyeballs at 3 dpf. Scale bar = 100 mm. (**B**–**D**) Statistical analysis of axial length and ocular area. (**E**,**F**) VMR testing demonstrates significant recovery of both ON and OFF responses in slc4a7-deficient zebrafish compensated with mRNA compared to those without compensation. *n* = 20 in each group. Rescue experiments were repeated three times. Bar plots represent the mean ± s.e.m. Data were analyzed using one-way ANOVA followed by Tukey’s post hoc test, *** *p* < 0.001, **** *p* < 0.0001, indicate significant differences from the *slc4a7* 1.00 ng group. ns means no significance. The scattered circles of various colors in the bar chart denote individual values of zebrafish eye-related parameters for different groups.

**Table 1 ijms-25-09613-t001:** Top 15 upregulated and downregulated genes in slc4a7 MO 1.00 ng eyeballs compared to control MO 1.00 ng eyeballs.

Top 15 Upregulated Genes				Top 15 Downregulated Genes			
Symbol	Log_2_(FC)	*p*-Value	Description	Symbol	Log_2_(FC)	*p*-Value	Description
gh1	10.00032086	1.17 × 10^−16^	growth hormone 1	fabp1a	−7.406957321	5.82 × 10^−8^	fatty acid binding protein 1a liver
urp2	8.86578455	2.29 × 10^−13^	urotensin II-related peptide	mc3r	−7.296820264	1.73 × 10^−8^	melanocortin 3 receptor
hoxb3a	8.557189431	2.53 × 10^−16^	homeobox B3a	guca1e	−7.207802662	2.18 × 10^−30^	guanylate cyclase activator 1e
hoxd3a	8.258759291	6.80 × 10^−11^	homeobox D3a	ghrhr2	−6.988088994	1.08 × 10^−6^	growth hormone releasing hormone receptor 2
gbx1	8.144372525	5.47 × 10^−11^	gastrulation brain homeobox 1	esrrgb	−6.917004252	4.07 × 10^−25^	estrogen-related receptor gamma b
ghrh	7.992290438	8.89 × 10^−11^	growth hormone releasing hormone	clul1	−6.740903422	0	clusterin-like 1 (retinal)
cmlc1	7.746139001	4.05 × 10^−9^	cardiac myosin light chain-1	cplx4b	−6.683814599	7.50 × 10^−6^	complexin 4b
hoxc4a	7.458320568	3.12 × 10^−9^	homeobox C4a	sult3st4	−6.221469929	1.09 × 10^−4^	sulfotransferase family 3, cytosolic sulfotransferase 4
lbx1b	7.345285686	6.31 × 10^−8^	ladybird homeobox 1b	slc1a8b	−6.003244066	4.55 × 10^−64^	solute carrier family 1 member 8b
c16h2orf66	7.326961637	1.39 × 10^−11^	chromosome 16 C2orf66 homolog	csf3r	−6.00266695	3.4 × 10^−4^	colony stimulating factor 3 receptor (granulocyte)
npvf	7.063882424	3.18 × 10^−15^	neuropeptide VF precursor	tas2r200.2	−5.870867913	8.82 × 10^−5^	taste receptor, type 2, member 200, tandem duplicate 2
evx1	6.985297952	5.68 × 10^−8^	even-skipped homeobox 1	mnx1	−5.839648876	1.36 × 10^−3^	motor neuron and pancreas homeobox 1
otpb	6.93493952	1.74 × 10^−89^	orthopedia homeobox b	scpp7	−5.730196353	1.62 × 10^−3^	secretory calcium-binding phosphoprotein 7
pomca	6.77243724	1.20 × 10^−4^	proopiomelanocortin a	cabp1b	−5.448790301	8.87 × 10^−59^	calcium binding protein 1b
hoxc5a	6.77205525	3.76 × 10^−6^	homeobox C5a	ctrl	−5.430447413	2.98 × 10^−3^	chymotrypsin-like

## Data Availability

The data that support the findings of this study are available from the corresponding author upon reasonable request.

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
