# Peer review of "Slc4a7 Regulates Retina Development in Zebrafish"

_ijms, 2024, doi:10.3390/ijms25179613_

Round 1

Reviewer 1 Report

Comments and Suggestions for Authors

This paper shows that knockdown of SLC4A7 in zebrafish has strong, negative effects on both structural and functional retinal development, contributing to shed light on the mechanisms involving SLC4A7 function. These observations may significantly contribute to the understanding of the involvement of this gene in inherited retinal degenerations. Overall, the manuscript is well-organized and the results are very clear, supporting the conclusions. A few things should be revised:

Line 38: The RetNet current address is  https://retnet.org

Paragraph 2.1: There are data about SLC4A7 expression in human ocular tissues and human retinal cell types, but there is no indication of the methods used to obtain these data. Please describe the source of the human tissues and define “normalized expression”. Also “relative expression”: relative to what?

Figure 1B: Different clusters can be clearly seen in this figure, but it is difficult to understand what they are if no labels are provided.

Line 83, “slc4a7 is predominantly situated in the plasma membrane”: slc4a7 is the gene as indicated in zebrafish. Since this sentence refers to humans, it should be SLC4A7. In addition, it is not correct to say that SLC4A7 (the gene) is localized to the plasma membrane: of course, it is the protein that derives from the expression of the gene.

Line 90: Here, the form slc4a7 is used to indicate the gene both in humans and in zebrafish, while on line 76 the form with capital letters (SLC4A7) was used. I do not know which one is preferable, but the form should be consistent.

Figure 2A: The red and blue dashed lines are only barely visible.

Lines 125 and 126: please define abbreviations RCVRN and AB.

Lines 138-139, “further activates the repair 138 mechanisms of Müller cells”: I don’t think these data demonstrate the activation of repair mechanisms. I would say that they induce Muller cell activation or reactive gliosis.

Line 162, “Supplementary Figure 2B-G”: there are only panels A and B in supplementary figure 2.

Line 169: There is no panel C in fig. 4.

Line 171: Figure 4B does not show TUNEL staining in retinal layers. In addition, it is not necessary to specify that TUNEL labeled cells are in GCL, INL, and ONL.

Figure 4B: The graph is supposed to show the number of TUNEL-positive cells, but judging from fig. 4A, it is impossible to detect and count single cells in the image of TUNEL staining (which also appears to be out of focus). It is evident that the control does not have any TUNEL staining, while most of the cells in the slc4a7 MO are labeled, therefore a quantitative analysis is not needed.

Line 262, “The immunostaining results demonstrate a marked decrease in fluorescence signals in photoreceptors, RPE, and amacrine cells …”: The mcherry fluorescence is not the result of an immunostaining procedure. Please also check other parts of the manuscript for this mistake.

Line 300, “Our immunostaining results …”: As observed above, these are not immunofluorescence data.

Comments on the Quality of English Language

English language is ok, but I suggest a thorough revision.

Reviewer 2 Report

Comments and Suggestions for Authors

The manuscript by Zhuang et al., describes a morpholino based study to demonstrate ocular defects resulting from slc4a7 reduction in zebrafish. In humans SLC4A7 have been defined in patients with various photoreceptor-based visual defects.  They demonstrate smaller eyes, increased cell death, and loss of photoreceptors as well as other cells in the eye. However, the use of a single morpholino to generate all results is problematic. The authors should refer the paper by [Stanier et al., PLoS Genet. 2017 Oct 19;13(10)] for guidelines for using morpholinos. Specific comments relating to this are:

1)        Not all phenotypes are rescued with mRNA injection. Eye size and VMR are rescued. Additional phenotypes shown in the paper should be rescued with mRNA to make this data convincing.  Furthermore, the authors state on line 242 that the morpholino was co-injected with mRNA. This is not an ideal way to perform the experiment, as simply injecting less of this mixture can lead to a reduced phenotype that would be interpreted as a rescue, when in fact, was simply less of a knockdown.  Ideally, the morpholino would be injected, then embryos divided in half, with one batch then being injected with the WT mRNA, then the second batch injected with buffer or embryo media. The person in injecting should be blinded to what is in each tube.

2)        Morpholinos often hade off target effects, as well as non-specific toxicity that can activate the p53 pathway and result in apoptosis. The author’s immunohistochemistry and RNA sequencing demonstrates increased apoptosis and upregulation of the p53 pathway. These is no way of knowing if this is specific to the morpholino or due to off-target effects/toxicity. 

3)        The generation of F0 crispants, or a stable mutant line of Slc4a7 would be much more convincing. 

Before publication, this reviewer highly recommends additional data to make the paper more convincing. This can include:

-              Additional rescue experiments using slc4a7 mRNA, performed as described above. 

-              The replication of at least some of the paper’s data in F0 crispants. 

Comments on the Quality of English Language

English language is fine. only minor editing needed. 

Reviewer 3 Report

Comments and Suggestions for Authors

I found this manuscript well-written and organized, based on a huge amount of analyses and results.

I suggest reporting the scientific name of the studied species with the eponym in the title and avoiding the use of words already reported in the title among the keywords.

The methods of SLC4A7 expression in humans, relative to results shown in paragraph 2.1 were not treated within the manuscript. This may confuse the readers.

Avoid in the results section comments as reported in lines 89-91, indeed these represent discussions. Please address accordingly the entire results section.

Best regards

Round 2

Reviewer 1 Report

Comments and Suggestions for Authors

Thank you for considering my comments and for your answers. I still have a couple of minor suggestions:

-          In the abstract, line 23, the reader cannot understand what type of “fluorescence observations” have been made unless you say that you have used mcherry to label photoreceptors and RPE. Something similar should be considered also in the discussion, line 256.

-           Figure 3. The title of the figure caption says “Immunofluorescence staining of the retina”, however only panel A reports immunofluorescence data.

Reviewer 2 Report

Comments and Suggestions for Authors

While the article is much improved with increased usage of rescue experiments, there are still concerns using a single morpholino to generate all results in the paper. This is no longer the standard in zebrafish field, even with mRNA rescue. A morpholino should be validated using stable mutant lines, or at least F0 crispants. F0 crispants can be generated in the same time frame and with similar costs to morpholino analysis. Please find a useful illustration of morpholino validation in the following paper (Stainer et al., 2017, PMID: 29049395). CRISPR interference could also be used to demonstrate a similar phenotype with similar time and cost considerations as morpholino analysis. Before recommending publication, demonstration of a similar ocular phenotype using genetic mutants (F0 crispants or stable mutant lines) or using an additional technique such as CRISPR interference is required. 
